# Structure shows that the BIR2 domain of E3 ligase XIAP binds across the RIPK2 kinase dimer interface

Mathilde Lethier[1], Karine Huard[1], Michael Hons[1], Adrien Favier[2,3,4], Bernhard Brutscher[2,3,4], Elisabetta Boeri Erba[2,3,4], Derek W Abbott[5], Stephen Cusack[1], Erika Pellegrini[1]

RIPK2 is an essential adaptor for NOD signalling and its kinase domain is a drug target for NOD-related diseases, such as inflammatory bowel disease. However, recent work indicates that the phosphorylation activity of RIPK2 is dispensable for signalling and that inhibitors of both RIPK2 activity and RIPK2 ubiquitination prevent the essential interaction between RIPK2 and the BIR2 domain of XIAP, the key RIPK2 ubiquitin E3 ligase. Moreover, XIAP BIR2 antagonists also block this interaction. To reveal the molecular mechanisms involved, we combined native mass spectrometry, NMR, and cryo-electron microscopy to determine the structure of the RIPK2 kinase BIR2 domain complex and validated the interface with in cellulo assays. The structure shows that BIR2 binds across the RIPK2 kinase antiparallel dimer and provides an explanation for both inhibitory mechanisms. It also highlights why phosphorylation of the kinase activation loop is dispensable for signalling while revealing the structural role of RIPK2–K209 residue in the RIPK2–XIAP BIR2 interaction. Our results clarify the features of the RIPK2 conformation essential for its role as a scaffold protein for ubiquitination.

## Introduction

The innate immune system is a frontline defence strategy that allows vertebrates to combat infectious agents. Pattern recognition receptors that recognise non-self, molecular patterns and their signalling pathways are key elements of the innate immune system. Nucleotide-binding oligomerization domain-containing 1 and 2 (NOD1 and NOD2) proteins are intracellular pattern recognition receptors sensing bacterial infections and contributing to gastrointestinal homeostasis (Philpott et al, 2014; Kayama Hisako, 2020; Trindade & Chen, 2020). Receptor activation by bacterial peptidoglycan breakdown products, γ-D-Glu-mdiaminopimelic acid (iE-DAP) or muramyl-dipeptide (MDP), respectively (Chamaillard et al, 2003; Girardin et al, 2003, 2016; Inohara et al, 2003), triggers

production of NF-κB and MAPK cascade activation. This results in transcriptional up-regulation and release of pro-inflammatory cytokines and antimicrobial peptides (Ferrand & Ferrero, 2013; Boyle et al, 2014; Philpott et al, 2014; Trindade & Chen, 2020). Alternatively, NOD signalling can be triggered by ER stress and small Rho GTPase (Keestra & Bäumler, 2014; Keestra-Gounder et al, 2016). NOD activation also triggers autophagy, through the recruitment of autophagy-related 16-like 1 (ATG16L1) (Travassos et al, 2010).

Dysregulated NOD signalling, which results in increased production of pro-inflammatory cytokines, impaired autophagy, and therefore chronic inflammation, is associated with several complex multifactorial inflammatory diseases, such as inflammatory bowel disease (IBDs), Blau syndrome, and early-onset sarcoidosis. IBDs, which include Crohn's disease (CD) and ulcerative colitis (UC), are a major social health problem in need of novel treatments (Kaplan & Windsor, 2021). Loss of function by mutations of NOD2 or ATG16L1 and deregulation of ubiquitination/deubiquitination in the NOD pathway can result in a genetic susceptibility to develop IBDs (Inohara et al, 2003; Cho & Abraham, 2007; Kuballa et al, 2008; Cooney et al, 2010; Philpott et al, 2014; Ananthakrishnan, 2015; Caruso et al, 2020; Sámano-Sánchez & Gibson, 2020; Honjo et al, 2021a; Zou et al, 2021). Dysregulation of NOD signalling has also been associated with other inflammatory disorders, for example, inflammatory arthritis, asthma, colorectal cancer, multiple sclerosis, and type 2 diabetes mellitus (Amar et al, 2011; Caruso et al, 2014; Philpott et al, 2014). For all these diseases, inhibition of NF-κB activation and therefore of pro-inflammatory cytokine production, has been suggested as a possible treatment (Atreya et al, 2008; Boyle et al, 2014; Philpott et al, 2014; Canning et al, 2015; Trindade & Chen, 2020).

Activation of the NOD receptors leads to the recruitment of receptor-interacting serine/threonine/tyrosine-protein kinase 2 (RIPK2) (Park et al, 2007; Boyle et al, 2014). RIPK2 is an adaptor protein, which comprises a kinase and a caspase activation and recruitment domain (CARD) connected by a disordered intermediate loop (Humphries et al, 2015). RIPK2 recruitment to activated

---

[1]European Molecular Biology Laboratory, Grenoble, France    [2]University Grenoble Alpes, IBS, Grenoble, France    [3]CNRS, IBS, Grenoble, France    [4]CEA, IBS, Grenoble, France    [5]Department of Pathology, Case Western Reserve University, Cleveland, OH, USA

Correspondence: epellegr@embl.fr; cusack@embl.fr
Mathilde Lethier's present address is ALPX, Grenoble, France

NOD receptor via CARD–CARD interactions, triggers RIPK2 auto-phosphorylation, filament formation (Pellegrini et al, 2018), and both Lys63-Ub and Met1-Ub ubiquitination. Ubiquitination is mediated by different E3 ligases including inhibitors of apoptosis (IAPs) and the linear ubiquitin chain complex (LUBAC) (Chin et al, 2002; Hasegawa et al, 2008; Bertrand et al, 2009; Tao et al, 2009; Damgaard et al, 2012; Boyle et al, 2014; Goncharov et al, 2018; Heim et al, 2020). Phosphorylation occurs at S176 (Dorsch et al, 2006) and Y474 (Tigno-Aranjuez et al, 2010), whereas Lys63-Ub has been reported at multiple sites (e.g., K182, K203, K209, K306, K326, K369, K410, K527, K537, and K538) (Hasegawa et al, 2008; Goncharov et al, 2018; Heim et al, 2020). Among the IAPs, the X-chromosome-linked inhibitor of apoptosis, XIAP (also known as inhibitor of apoptosis protein 3 [IAP3] and baculoviral IAP repeat-containing protein 4 [BIRC4]) is thought to be the most critical ubiquitin ligase in the NOD2-RIPK2 signalling pathway (Krieg et al, 2009; Bertrand et al, 2011; Damgaard et al, 2012; Stafford et al, 2018). Similarly to the other IAP proteins, it comprises 3 baculovirus IAP repeat (BIR) domains (BIR1, BIR2, and BIR3) and a C-terminal RING domain (Mace et al, 2010). XIAP binds to the kinase domain of RIPK2 through its BIR2 domain (Krieg et al, 2009; Damgaard et al, 2013; Goncharov et al, 2018; Hrdinka et al, 2018). It then coordinates the conjugation of Lys63-Ub chains (originally thought to be at K209, but now considered to be K410 and K538, see the Discussion section) and triggers the recruitment of LUBAC, which in turn promotes Met1-Ub conjugation and the recruitment of the IKK complex, an essential step for NF-κB activation (Damgaard et al, 2012; Hrdinka & Gyrd-Hansen, 2017; Goncharov et al, 2018). The absence of XIAP causes defective RIPK2 ubiquitination, defective recognition by effector proteins of ubiquitinated RIPK2 (e.g., LUBAC), reduced inflammatory signalling, and inadequate bacterial clearance (Bauler et al, 2008; Krieg et al, 2009; Damgaard et al, 2012, 2013; Speckmann et al, 2013). Mutations in the *XIAP* gene promote inflammatory pathologies, such as the X-linked lymphoproliferative disease type 2 (XLP-2) and also causes very early onset IBD (Damgaard et al, 2013; Pedersen et al, 2014; Nielsen & LaCasse, 2017). XLP-2 mutations on the XIAP BIR2 domain abrogate the interaction with RIPK2, and therefore the ubiquitination of RIPK2 (Damgaard et al, 2013). In the autophagy pathway, NOD2 recruits ATG16L1, which in turn binds the kinase domain of RIPK2 and negatively impact its ubiquitination to suppress NF-κB activation by Toll-like receptor 2 (Honjo et al, 2021a).

RIPK2 is an essential downstream component of the NOD and ATG16L1 signalling pathways. In these pathways, recruitment of RIPK2 to NOD1 or NOD2 triggers NF-κB activation. Therefore, the inhibition of RIPK2 kinase activity has been suggested and demonstrated in vivo to be beneficial as a therapeutic strategy for the inflammatory diseases cited above, in particular, for IBDs (Negroni et al, 2009; Philpott et al, 2014; Tigno-Aranjuez et al, 2014; Canning et al, 2015; Nachbur et al, 2015; Salla et al, 2018; Haile et al, 2019; Watanabe et al, 2019; Honjo et al, 2021b). However, the molecular significance of the connection between the kinase activity of RIPK2, and its role as a scaffold protein for ubiquitination and binding of downstream signalling molecules has remained unclear. Furthermore, several studies (Abbott et al, 2004; Windheim et al, 2007; Nachbur et al, 2015; Goncharov et al, 2018; Hrdinka et al, 2018) clearly show that RIPK2 auto-phosphorylation activity is not necessary to trigger NF-κB activation. It has also been found that RIPK2 is phosphorylated even in unstimulated cells, making the role of RIPK2 kinase activity even more cryptic (Heim et al, 2020).

In 2018, it was shown that the high nanomolar potency of certain RIPK2 kinase inhibitors (e.g., ponatinib, GSK583, and CSLP37/43) depends on these molecules preventing the interaction of RIPK2 with the XIAP BIR2 domain, and hence inhibiting RIPK2 ubiquitination, rather than their ability to inhibit RIPK2 kinase activity (Goncharov et al, 2018; Hrdinka et al, 2018). In agreement with previous work (Krieg et al, 2009; Damgaard et al, 2013), Goncharov et al (2018) also found that selective XIAP BIR2 antagonists interfere with the XIAP–RIPK2 interaction, blocking NOD2-mediated RIPK2 ubiquitination and subsequent activation of inflammatory signalling.

Here, we present the structure of RIPK2 kinase bound to the BIR2 domain of XIAP determined by single-particle cryo-electron microscopy (SPA cryo-EM) at a nominal resolution of 3.15 Å. Combined with biophysical characterisation and in cellulo validation of interaction interfaces, the structure provides a molecular explanation for the inhibitory mechanism of antagonists. Moreover, it explains why RIPK2 auto-phosphorylation is not required for the protein–protein interaction. The structure is consistent with biochemical data reported in the literature and provides new insight into the regulation of RIPK2 ubiquitination.

## Results

### In vitro reconstitution of the RIPK2–XIAP BIR2 complex and its stoichiometry

The interaction between RIPK2 and XIAP occurs between the kinase domain of RIPK2 (residues 1–300) and the BIR2 domain of XIAP (Krieg et al, 2009; Goncharov et al, 2018; Hrdinka et al, 2018). Previous interaction studies have used either the XIAP BIR2$_{AG}$$^{124–260}$ (Krieg et al, 2009) or the XIAP BIR2$_{AG}$$^{124–240}$ construct (Goncharov et al, 2018; Hrdinka et al, 2018). Here, the suffix AG denotes two mutations (C202A and C213G), which were originally made to obtain a suitable sample for structure determination by NMR spectroscopy and found to improve sample quality by limiting protein aggregation (Sun et al, 1999). Indeed, XIAP BIR2$_{AG}$$^{124–240}$ was used to determine the NMR structure of XIAP BIR2 (Sun et al, 1999), whereas the crystal structure was obtained using a shorter construct, XIAP BIR2$_{AG}$$^{154–240}$ (Lukacs et al, 2013). Both structures show that XIAP BIR2$_{AG}$$^{154–240}$ is long enough to encompass the folded BIR2 domain (residues 163–230, based on annotations in Uniprot entry P98170), whereas XIAP BIR2$_{AG}$$^{124–240}$ comprises a linker region (residues 124–154), known to be critical to inhibit caspase-3 and -7 activities (Chai et al, 2001; Riedl et al, 2001; Silke et al, 2001; Scott et al, 2005).

In vitro we reconstituted the RIPK2–XIAP BIR2 complex using either RIPK2$^{1–317}$ or RIPK2$^{1–300}$ and both XIAP BIR2$^{124–240}$ and XIAP BIR2$^{154–240}$ as demonstrated by size exclusion chromatography and we obtained similar results using the corresponding AG constructs (XIAP BIR2$_{AG}$$^{124–240}$ and XIAP BIR2$_{AG}$$^{124–240}$) (Figs 1A and B and S1A–D).

To determine the stoichiometry of these complexes, we used native mass spectrometry (MS) on the four samples. When the short

BIR2 construct was analysed, the main stoichiometry ratios were 2:1 and 2:2 (RIPK2:XIAP BIR2$^{154-240}$) with higher abundance for the 2:1 oligomer (Fig 1C, top left panel, Table S1). For the AG construct, the most abundant signals also corresponded to 2:1 and 2:2 ratios of the components (RIPK2:XIAP BIR2$_{AG}$$^{154-240}$), with higher abundance for the 2:2 oligomer (Fig 1C, top right panel, Table S1). When the longer construct was investigated, the main stoichiometry ratios were also 2:1 and 2:2 for RIPK2–XIAP BIR2$^{124-240}$ (Fig 1C, bottom left panel). The stoichiometry becomes 2:2 for the RIPK2–XIAP BIR2$_{AG}$$^{124-240}$ sample (Fig 1C, bottom right panel). These data clearly indicate that two molecules of RIPK2 are required for the interaction with XIAP BIR2. Moreover, the data highlight that the AG mutation affects the stoichiometry (Table S1).

**XIAP interacts with RIPK2 using exclusively the folded domain**

We next investigated whether the linker regions flanking the BIR2 domain contribute to RIPK2 binding. We designed several constructs comprising the BIR2 domain (XIAP BIR2$^{124-263}$, XIAP BIR2$^{93-240}$, and XIAP BIR2$^{93-263}$) (Fig 1A) and assessed their ability to bind to the kinase domain of RIPK2 by determining their dissociation constant (Kd) through Microscale Thermophoresis measurements. We found that the Kd was in the nM range (Kd= 85.6 ± 18.7 nM) for XIAP BIR2$^{154-240}$ and did not find any significant difference amongst the constructs (Fig S2A and B).

We then applied solution NMR spectroscopy to define the XIAP BIR2 residues that are involved in the interaction with RIPK2. We successfully produced uniformly $^{13}C/^{15}N$-labelled XIAP BIR2$_{AG}$$^{124-240}$. Backbone ($^{1}H$, $^{15}N$ and $^{13}C$) NMR assignments of XIAP BIR2$_{AG}$$^{124-240}$ were obtained from a series of three-dimensional HNC-type correlation experiments. Fig 2A shows a superposition of the amide $^{1}H$-$^{15}N$ correlation spectra recorded for isolated XIAP BIR2$_{AG}$$^{124-240}$ (black), and in complex with unlabelled RIPK2$^{1-317}$ (red). Only residues in the N- and C-terminal extensions of the XIAP BIR2 construct that are highly flexible in solution (Fig 2C) remain observable in the complex, whereas NMR peak intensities for residues in the folded part are significantly attenuated because of the relatively large particle size of the RIPK2–XIAP BIR2 complex (Fig 2). These data clearly show that only the folded BIR2 domain is involved in the interaction, whereas both N- and C-terminal extensions remain highly mobile (Fig 2B).

Furthermore, NMR-detected translational diffusion measurements indicate that upon interaction with RIPK2, the apparent molecular size is increased by a factor of about 5.7 (Fig S3). This is in good agreement with the presence of a RIPK2 kinase dimer in the complex, and a 1:2 RIPK2:BIR2 stoichiometry, although the presence of 2:2 RIPK2:BIR2 complexes cannot be excluded. Based on the results obtained so far, we decided to use only the short BIR2 constructs (residues 154–240) for structural studies.

**Structure determination by cryo-EM of RIPK2–XIAP BIR2 complex**

Exhaustive crystallisation trials failed to produce crystals containing either the RIPK2$^{1-317}$-XIAP BIR2$^{154-240}$ or the RIPK2$^{1-317}$-XIAP BIR2$_{AG}$$^{154-240}$ complex, resulting only in RIPK2 dimer crystals. Therefore, we attempted to solve the structure of RIPK2$^{1-317}$-XIAP BIR2$^{154-240}$ by cryo-

EM. To make the complex, we purified both WT domains to homogeneity. We added ATP-MgCl$_2$ to the kinase sample to promote full phosphorylation (Pellegrini et al, 2017) and we combined it with an excess of XIAP BIR2 protein. The sample was further purified by size-exclusion chromatography and the most enriched complex fraction was selected for vitrification (Fig S4A and B). We performed data collections with a Titan Krios, combining untilted and tilted data. We obtained two-dimensional (2D) class averages showing structural features and finally obtained a map at a nominal resolution of 3.15 Å (Figs S4–S6 and Table S2). In this map, we were able to unambiguously dock two RIPK2 kinase molecules, arranged as an antiparallel dimer, as in RIPK2 crystal structures (PDB ID: 5NG0, 4C8B), bound to one molecule of XIAP BIR2 (PDB ID: 4J3Y) (Lukacs et al, 2013; Canning et al, 2015; Pellegrini et al, 2017) (Figs 3A–D and S7A and B and Table S2). We immediately observed that one of the BIR2 mutations (the C213G) known to improve protein quality is at the interaction interface (Fig 3F). Therefore, we did not pursue the structure determination of the RIPK2$^{1-317}$–XIAP BIR2$_{AG}$$^{154-240}$ complex.

Both kinase chains are in the active-like conformation based on their αC-helix position (IN conformation) and density can be observed for nucleotide in the active site (Figs 3A and S7C and D). However, the poorly resolved density around the αC-helix does not allow confident assignment of rotamers, in particular, that of Glu66. Therefore, we cannot confirm the presence of the salt bridge between Lys47 and Glu66, which is required for catalysis. The N-termini anti-parallel β-strands described to be present in the dimeric structure of active-like RIPK2 (Pellegrini et al, 2017), is absent (Fig S8A and B).

In parallel, we used Alphafold2 as implemented in Colabfold (Jumper et al, 2021; Evans et al, 2022 Preprint; Mirdita et al, 2022) to predict the structure of dimeric RIPK2 with bound XIAP BIR2. In agreement with our experimentally derived model, the prediction shows no anti-parallel β-strand interaction of the kinase N-termini. Interestingly, the predicted aligned error diagram shows low confidence interaction between the kinase and the BIR2 domain, despite the predicted model being very close to that observed (Fig S8C).

**The RIPK2–XIAP interaction requires RIPK2 kinase dimerization**

The final model of the complex corresponds to a mass of 84 kD (Table S1), with a 2:1 stoichiometry, in agreement with both native MS and translational diffusion measurements by NMR. The structure shows that XIAP BIR2 binds at the RIPK2 dimer interface, interacting with the C-lobe of one kinase molecule (Kinase_A) and the N-lobe of the other one (Kinase_B) (Fig 3A). The XIAP BIR2-binding site on RIPK2 comprises two regions that were already shown to be part of the XIAP BIR2-binding site through biochemical and cellular assays. These are the loop between strands β2 and β3 sheet or R36/R41 patch (Hrdina et al, 2018) and the so-called regulatory region formed by the αE-helix and the loop between helix αE and αEF-helix, which includes residues I208–K209 (Heim et al, 2020). RIPK2 αH-helix is also part of the binding site. Indeed, our RIPK2–XIAP BIR2 structure shows how these three kinase regions are arranged in the antiparallel kinase dimer to create a platform for XIAP BIR2 domain binding.

To confirm observed interactions, we mutated to either alanine or leucine the residues at the RIPK2–XIAP BIR2 interface. The

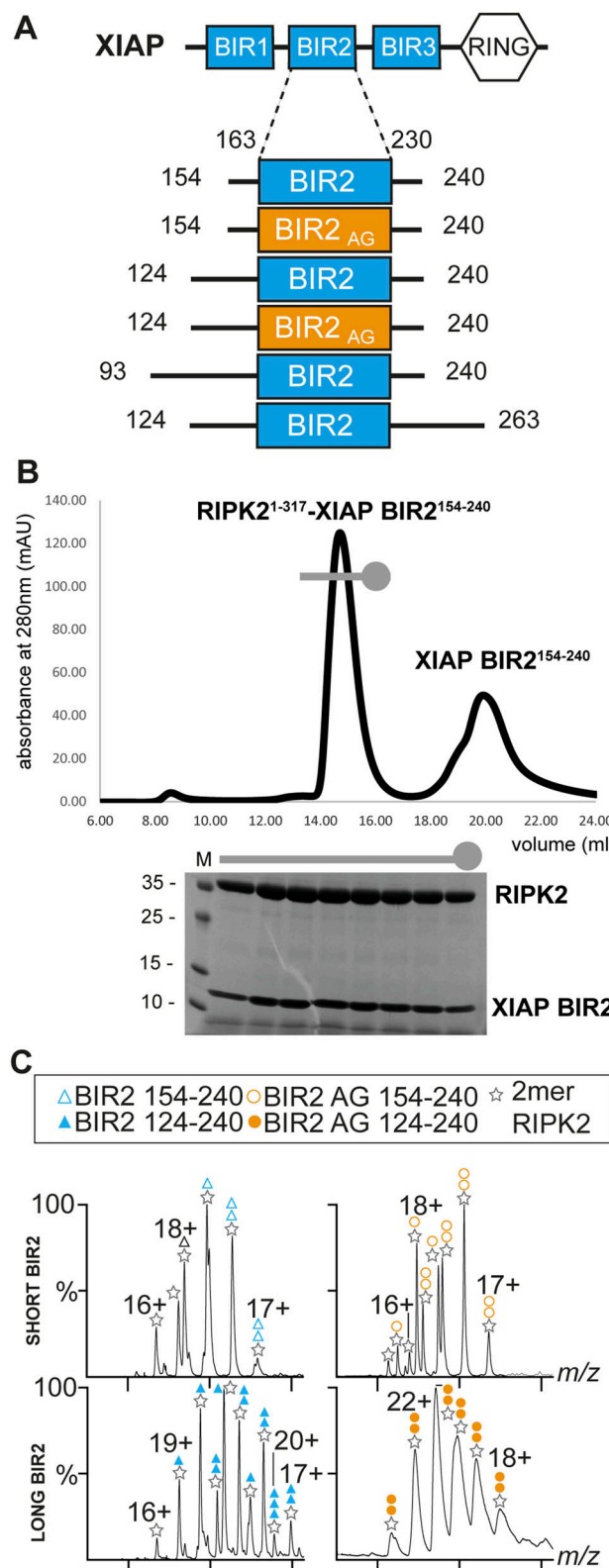

**Figure 1. In vitro reconstitution and stoichiometry of the RIPK2–XIAP BIR2 complex.**
**(A)** Overview of the XIAP BIR2 constructs used in this article. **(B)** SEC profile and SDS–PAGE gel of RIPK2$^{1–317}$–XIAP BIR2$^{154–240}$ complex. Uncropped SDS–PAGE gel is

mutations were performed in tagged full-length constructs, which were transiently expressed in mammalian cells. 22 h after adding the NOD2 activator MDP, cells were harvested and the effect of the mutations on complex formation was assessed by pull-downs (Fig 4A–C).

XIAP BIR2 uses loop 176–178 to mainly interact with Kinase_A and loop 209–214 to interact with Kinase_B (Fig 3E and F). The XIAP BIR2 loop 176–178, which includes Y176 and H178, makes electrostatic interaction with the R36/R41 patch. Pull-down data show that mutant R36L in RIPK2-Kinase_A reduces the complex interaction dramatically, whereas mutants R41L, R36L/R41L, and XIAP mutant Y176A abort complex formation completely. The structure shows that the two arginines also contribute to the kinase dimerization interface (Fig 3E). In the apo-structure (PDB ID: 5NG0, [Pellegrini et al, 2017]) R41 of Kinase_A is H-bonded to D291 of Kinase _B, whereas R36L makes polar contacts with waters located in between the two kinase molecules. Therefore, in the case of the arginine mutants, we cannot distinguish whether the mutation affects complex formation or kinase dimerization or both. XIAP H178 potentially makes a hydrogen bond with the oxygen group of D39, which is also part of the R36/R41 patch. However, the mutations XIAP H178A and RIPK2 D39L do not impair complex formation (Figs 3E and 4B and C).

The XIAP BIR2 loop 209–214 makes electrostatic interactions with the C-lobe of Kinase_B, which comprises residues belonging to the αH-helix (E279, S282, and K285) and to the regulatory region (K209 and I208). N209, E211, and D214 from XIAP BIR2 loop 209–214 form a negatively charged binding site that appears to lock Kinase_B residues K209–I208. Mutations of these XIAP loop residues blocks complex binding, except for C213A, which agrees with the fact that the XIAP BIR2$_{AG}$ construct can still bind the kinase (Fig 4C). Mutations of RIPK2 K209, E279, S282, and K285 destroy the binding, validating the interaction interface (Fig 4B). In the apo structure (PDB ID: 5NG0, [Pellegrini et al, 2017]), K209, E279, S282, K285 are not contributing to the kinase dimerization interface, which implies that their mutation likely affects BIR2 binding only. Indeed, recombinant RIPK2–K209R, RIPK2–K209A, and RIPK2–S282L retain the dimerization profile of RIPK2 wt on size-exclusion chromatography, but they do not bind XIAP BIR2 anymore (Fig S9A–D).

The kinase regulatory region also includes residues N137 and N133, which are located at both XIAP BIR2-binding site and dimerization interface (Fig 3F). N137L mutation decreases complex formation, whereas double mutation N137L/N133L aborts binding (Figs 3F and 4B and C). To test this in cellular assays, RIPK2–/–– immortalized bone marrow-derived macrophages (Chirieleison et al, 2016) were transduced with a doxycycline-inducible viral vector expressing WT RIPK2 or N137L RIPK2. Western blotting showed equivalent expression upon doxycycline induction (Fig 4D). Upon treatment with both doxycycline and MDP, both IRG1 and CXCL10 gene expression increased substantially in the WT RIPK2-

reported in Fig S1B. **(C)** Native mass spectrometry results for short BIR2 (top spectra) and long BIR2 (bottom spectra) constructs. The main stoichiometry ratios are 2:1 and 2:2 (RIPK2:XIAP BIR2), where two molecules of RIPK2 are required for the interaction with XIAP BIR2. The stoichiometry becomes 2:2 in the case of RIPK2- XIAP BIR2$_{AG}$$^{124–240}$.

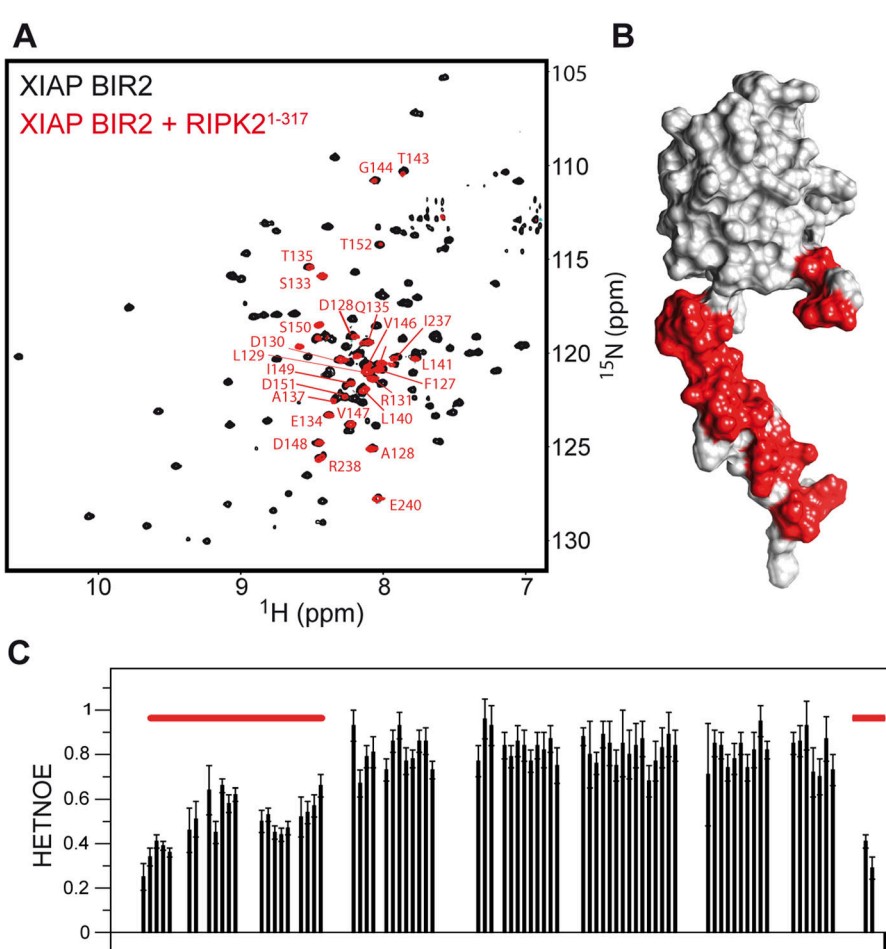

**Figure 2. NMR characterization of the XIAP BIR2 interaction with RIPK2 kinase in solution.**
**(A)** Overlap of $^1$H-$^{15}$N BEST-TROSY correlation spectra (25°C) recorded at 850 MHz $^1$H frequency on samples of $^{15}$N-labeled XIAP BIR2 (black) and a 1:1 complex of $^{15}$N-labeled XIAP BIR2 and unlabeled RIPK2$^{1–317}$ (red). BIR2 residues that remain visible in the complex are annotated by their amino acid type (one-letter code) and residue number. No or only small chemical shift changes are detected for these residues between the free BIR2 protein and the complex, indicating that these protein segments are not involved in the interaction. **(B)** Surface representation of the XIAP BIR2 structure (PDB ID: 1C9Q, Sun et al, 1999). The N- and C-terminal protein segments that are not involved in the interaction with RIPK2 are highlighted in red. **(C)** {$^1$H}-$^{15}$N heteronuclear NOE (HETNOE) ratios measured for XIAP BIR2 at 25°C and 850 MHz $^1$H frequency. The protein segments that remain visible in the NMR correlation spectrum of (A) upon interaction with RIPK2 are highlighted by red bars. These N- and C-terminal segments show reduced HETNOE ratios (≤0.6), indicative of significant fast (sub-ns) time scale backbone mobility, whereas for the central part, an average HETNOE of 0.8 is measured, in agreement with a globular protein domain. The flexibility of the N- and C-terminal segments is preserved in the complex, which makes them NMR-observable despite their high molecular weight.

## Discussion

In crystal structures, the kinase domain of RIPK2 is arranged in an antiparallel dimer (Canning et al, 2015; Charnley et al, 2015; Pellegrini et al, 2017). We previously showed that RIPK2 is also in a dimeric state in solution suggesting that kinase activation is coupled to dimerization (Pellegrini et al, 2017). This is also true for the D146N mutant, which corresponds to a protein without kinase activity, yet still able to trigger NF-κB activation (Goncharov et al, 2018; Hrdinka et al, 2018). In the case of the K47R (or K47A) dead kinase, NF-κB activation is still possible and the protein is both dimeric and monomeric in solution (Pellegrini et al, 2017; Goncharov et al, 2018; Hrdinka et al, 2018). Hitherto, a rationale for the RIPK2 dimer in NOD signalling has not been identified.

reconstituted cells. However, this response was significantly blunted in the N137L RIPK2-reconstituted cells (Fig 4E), indicating that this interface is crucial for downstream RIPK2-driven gene expression.

The cryo-EM reconstruction of the RIPK2$^{1–317}$-XIAP BIR2$^{154–240}$ complex shows the RIPK2 as a similar antiparallel dimer to that seen in crystal structures. Consistent with our NMR results, only the folded BIR2 domain binds to RIPK2. Importantly, the bound XIAP BIR2 domain engages both monomers with two different loops, highlighting the importance of the RIPK2 dimer as a scaffold for downstream ubiquitination. The resulting interface is different from the one described for the caspase 3 and 7-binding sites (PDB IDs: 1I3O, 1I51 and Fig S10A and B) (Chai et al, 2001; Riedl et al, 2001). All the residues previously described to disrupt the RIPK2–XIAP BIR2 interaction contribute directly to protein–protein interaction, except I212, which is located deeper in the Kinase B structure (Damgaard et al, 2013; Cavallari et al, 2017; Hrdinka & Gyrd-Hansen, 2017). In particular, the two RIPK2 patches suggested to be part of the interaction, the R36/R41 basic patch (Hrdinka et al, 2018) and the regulatory region comprising K209 (Heim et al, 2020), which are 40 Å apart in the monomer, are in fact close to each other in the dimer because of the antiparallel kinase arrangement. Our structure together with biochemical and cellular data describe how these two regions, together with the kinase αH-helix, form a

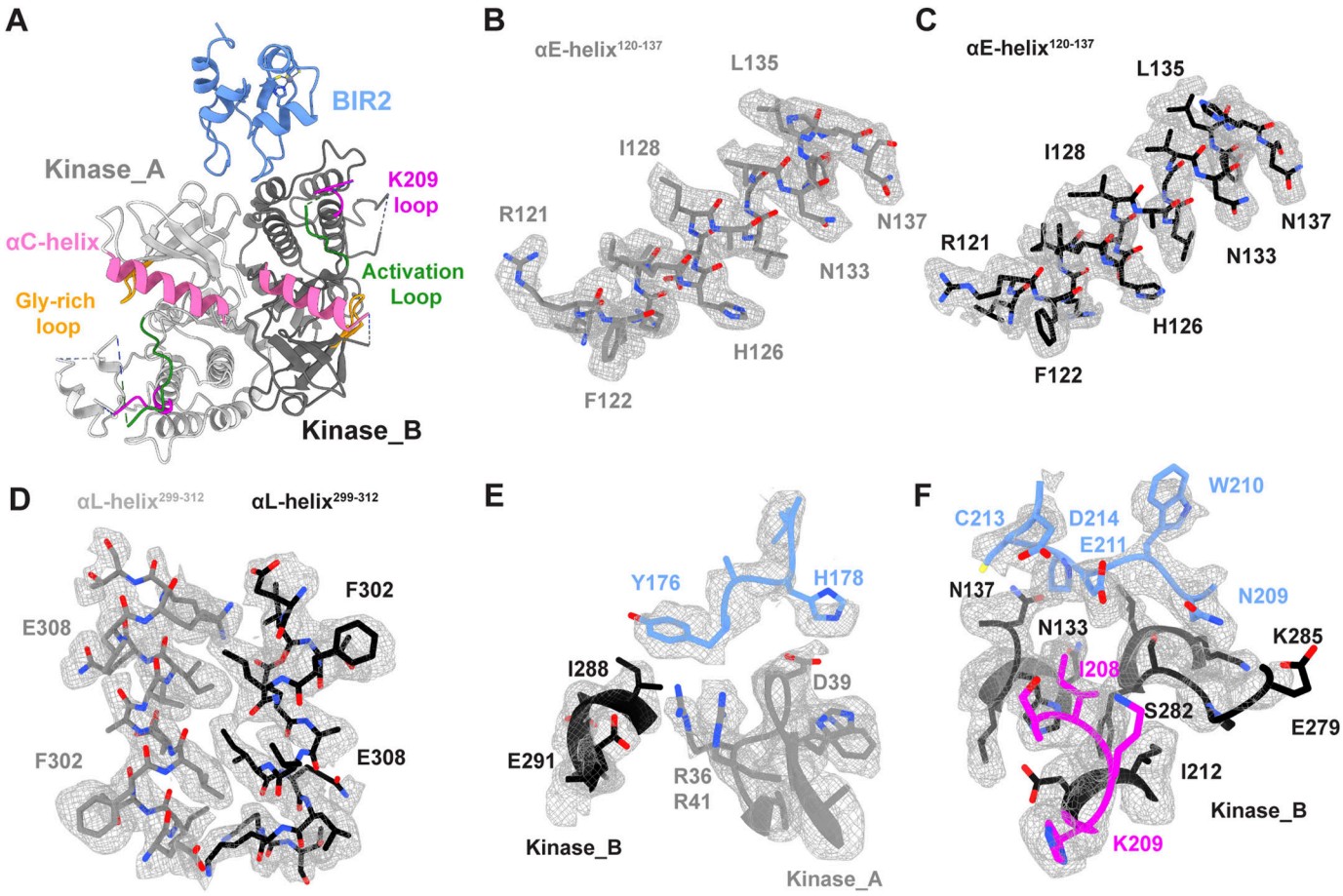

**Figure 3. Structure of the RIPK2$^{1-317}$–XIAP BIR2$^{154-240}$ complex.**
**(A)** Ribbon representation of the RIPK2$^{1-317}$-XIAP BIR2$^{154-240}$ structure. XIAP BIR2 is colored in light blue, kinases molecules in light and dark grey (Kinase_A, Kinase_B), αC-helix in pink (residues 52–72), Gly-rich loop in yellow, activation loop in green, and K209 loop in magenta (as described in Pellegrini et al, 2017). **(B, C, D, E, F)** Assignment according to the cryo-EM density features of (B) Kinase_A αE-helix residues (120–137), (C) Kinase_B αE-helix residues (120–137), (D) Kinase_A and Kinase_B αL-helices and their interaction (residues 299–312), (E) XIAP BIR2 interaction with Kinase_A, (F) XIAP BIR2 interaction with Kinase_B.

unique binding site for the XIAP BIR2 domain on the RIPK2 kinase dimer.

No interaction is observed between the XIAP BIR2 domain and the kinase activation loop, which indicates that XIAP binding requires RIPK2 to be in dimeric and αC-helix IN conformation, but not necessarily phosphorylated on the activation loop. This agrees with the fact that WT, the D146N mutant, and the K47R mutant, which all form dimers even though with different stabilities, can all trigger NF-κB signalling (Goncharov et al, 2018; Hrdinka et al, 2018). Our structure supports the hypothesis that the phosphorylation state is not relevant for NF-κB activation in NOD signalling.

Native MS data show both 2:1 and 2:2 stoichiometry, whereas NMR translational diffusion measurements suggest a 2:1 stoichiometry. We found these data quite surprising, as we expected two XIAP BIR2-binding sites based on the symmetrical arrangement of the kinase dimer. In agreement with the biophysical data, we could not identify any 2D classes with double XIAP BIR2, suggesting that the presence of a second copy of XIAP BIR2 in the RIPK2- XIAP BIR2$_{AG}$$^{124-240}$ sample might be related to the ability of XIAP BIR2 domain to homodimerise (Lukacs et al, 2013). Interestingly, the complex

reconstruction does not show any anti-parallel β-sheet interaction of the kinase N-termini, which appear to be flexible. Moreover, the two kinase monomers are closer in comparison to the original active structure (PDB ID: 5NG0, Fig S8A and B) (Pellegrini et al, 2017). From these observations, we speculate that binding of XIAP BIR2 slightly perturbs the kinase dimer, which might be the reason for which only one XIAP BIR2 molecule can be bound.

Based on the RIPK2–XIAP BIR2 structure here described, we can now provide a molecular explanation for the mechanism of XIAP BIR2 and RIPK2 inhibitors that prevent complex formation. Several articles reported that SMAC mimetic compounds, which bind in the so-called BIR IBM (IAP-binding motif)-binding groove, abolish the XIAP BIR2 interaction with RIPK2 resulting in its impaired ubiquitination and diminished NF-κB signalling (Krieg et al, 2009; Damgaard et al, 2013; Goncharov et al, 2018). These compounds mimic the IBM, a four-residue linear motif (AVPI) belonging to the SMAC/DIABLO protein (second mitochondria-derived activator of caspases/direct IAP-binding protein with low pI), which antagonises the binding of caspases to IAP proteins (Gyrd-Hansen & Meier, 2010). The XIAP BIR2 domain binds to the IBM of activated caspase-3 and caspase-7 (Scott

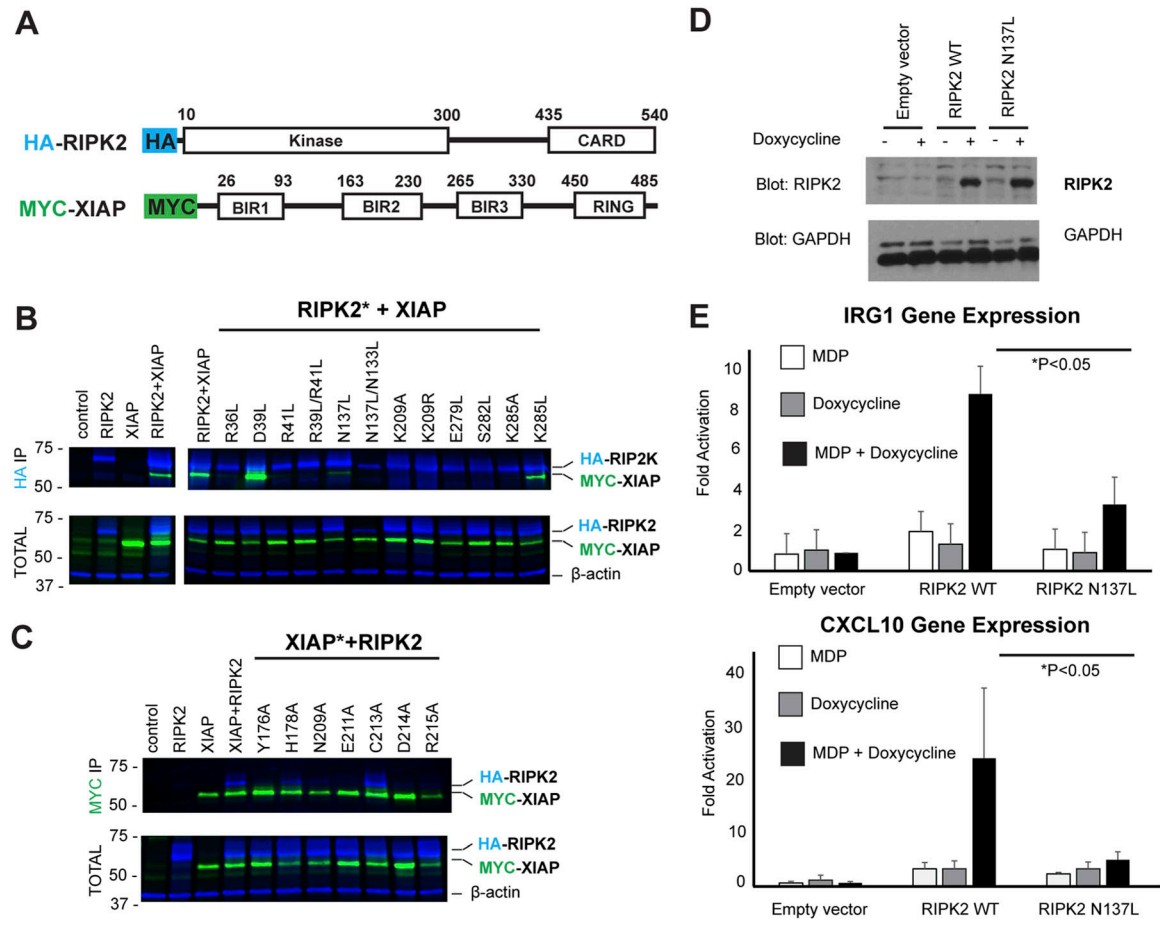

**Figure 4. Validation of the interaction interfaces between RIPK2 and XIAP BIR2.**

**(A)** Schematic representation of the RIPK2 and XIAP constructs used for the expression of wt and mutant proteins in HEKT293 cells. **(B, C)** Results of expression and pull down (IP, immunoprecipitation) on HA-RIPK2 and MYC-XIAP proteins from HEKT293 cells. Lines corresponding to IPs against HA and β-actin are in blue, whereas lanes corresponding to IP against MYC re in green. IPs have been repeated twice. **(D, E)** Reconstitution of RIPK2⁻/⁻ iBMDMs with doxycycline-inducible forms of RIPK2. **(D)** Cells were transduced with doxycyline-inducible lentiviral vectors expressing WT or N137L RIPK2. After selection, clones were pooled. They were then left untreated or treated overnight with 500 ng/ml doxycycline. Western blotting was performed and showed equivalently inducible levels of RIPK2 WT and RIPK2 N137L. **(E)** The cells generated in Panel A were then left untreated or were treated with 500 ng/ml doxycycline overnight. After this treatment, cells were either not exposed or exposed to 10 mg/ml MDP for 4 h qRT–PCR was then performed using the NOD2-inducible IRG1 and CXCL10 genes. Biological triplicates, each with two technical replicates, are shown. Source data are available for this figure.

et al, 2005). Comparison between our structure and that of XIAP BIR2 in complex with peptide AVPI (PDB ID: 4J46) (Lukacs et al, 2013) shows that the BIR2 inhibitor prevents complex formation with RIPK2 occluding the IBM groove (Fig S10C). Indeed the BIR2 inhibitor engages polar contacts with residues of the XIAP loop 209–214, which we have here described to guide the interaction between the BIR2 domain and the C-lobe of Kinase_B. This structural observation is in agreement with the fact that XLP-2 BIR2 mutations that either destabilise the domain or belong to the IBM groove abolish the RIPK2–BIR2 interaction (Sun et al, 1999; Damgaard et al, 2013; Lukacs et al, 2013).

Hrdinka et al (2018) observed that RIPK2 inhibitors that affect both kinase activity and ubiquitination state (e.g., ponatinib, GSK583, CSLP37/43, and 7f) interact with residues of the so-called back pocket of the kinase (A45, K47, I96, and gatekeeper T95), a binding site that becomes accessible when the activation loop is in the inactive conformation. They further speculated that binding

of inhibitors to this pocket could induce conformational changes at the basic patch surface exposed on the top of it, comprising R36 and R41. Single and double mutations of these two arginines impair the interaction with the XIAP BIR2 domain (Fig 4B), (Hrdinka et al, 2018). Our structure shows that these residues are involved in the RIPK2–BIR2 interaction, and they are positioned at the kinase dimerization interface (Fig 3E). Indeed, any perturbation of this zone might prevent binding between the two proteins by affecting either the binding itself or by destabilising the kinase dimer.

To provide a structural explanation for the ability of these compounds to prevent complex formation, we compared the crystal structures of RIPK2 in complex with ponatinib, GSK583 or CSLP18, with the RIPK2-XIAP BIR2 cryo-EM structure. No conformational difference exists at the XIAP BIR2-binding site, whereas in the active site, the inhibitor binding perturbs the DFG-in conformation and disorders the activation loop beyond residue

164 (PDB IDs: 4C8B, 5J7B and 6FU5, Fig S11A–D) (Canning et al, 2015; Haile et al, 2016; He et al, 2017). In the active conformation residues, 164–171 can still be traced, suggesting that the ability of inhibitors to completely destabilise the activation loop and the DFG-in conformation might contribute to their capacity to prevent complex formation.

The RIPK2–XIAP BIR2 structure confirms that RIPK2 K209 residue is part of the protein–protein interface (Fig 3F). K209 was originally proposed as a ubiquitination site, based on the observation that the K209R mutant blocks RIPK2 ubiquitination and it has a loss of function phenotype when overexpressed with ubiquitin (Hasegawa et al, 2008; Tigno-Aranjuez et al, 2013). However, two recent studies, which used a proteomic approach to determine RIPK2 ubiquitination sites by XIAP, failed to detect ubiquitination at K209 and instead suggested that the relevant lysines were K410 and K538 (Goncharov et al, 2018; Heim et al, 2020). Moreover, our biochemical and pull-down data, in agreement with published data (Heim et al, 2020), show that mutants K209A and K209R block the RIPK2–BIR2 interaction (Figs 4B and S9A–D).

Alternatively, K209 could be an ubiquitination site targeted by other IAPs, such as cIAP1 and 2 (Bertrand et al, 2009, 2011), even though it is still controversial whether these cIAPs contribute to NOD signalling by ubiquitination of RIPK2 (Stafford et al, 2018). BIR domains share a common fold, and their ability to interact with different partners depends on sequence differences at surface-exposed positions (Budhidarmo & Day, 2015). Comparison of the crystallographic structure of XIAP BIR2 domain with the cIAP1 and two BIR2 domain, predicted using Alphafold2, shows high structural conservation for the IBM groove (Fig S12B). Indeed, sequences of the BIR2 domains of XIAP, cIAP1, and cIAP2 show high conservation in the XIAP loop 200–214 (Fig S12A). Our pulldown assay shows that single mutants of XIAP residues N209, E211, and D214 in loop 200–214 abolish protein–protein interaction (Fig 4C). These observations suggest that the BIR2 domains of XIAP and cIAP1-2 are highly similar and they probably interact with RIPK2 dimer in a similar fashion, making K209 inaccessible for ubiquitination. Our results provide strong evidence that the importance of K209 in RIPK2 ubiquitination is related to its fundamental contribution to the binding of BIR2 domains.

In conclusion, the biophysical and structural characterisation of the RIPK2–XIAP BIR2 interaction described here reveals that the scaffold function of RIPK2 in binding XIAP requires it to be a dimer. This study also provides a molecular explanation to why the phosphorylation state of RIPK2 is not relevant for NOD signalling and why XIAP BIR2 antagonists and potent RIPK2 inhibitors abrogate RIPK2 ubiquitination and NF-κB activation by preventing complex formation. Moreover, considering the RIPK2–XIAP BIR2 structure together with our biochemical and previously published observations, we confirm that K209 is critical for RIPK2 ubiquitination most likely because it contributes to the RIPK2–XIAP BIR2 interaction interface. In conclusion, this study, by providing explanations for several previously suggested mechanisms, advances our knowledge on the scaffold role of the RIPK2 protein and its interaction with XIAP BIR2. These results might be of value for the design of more potent and specific RIPK2 and XIAP BIR2 inhibitors.

# Materials and Methods

## Protein constructs

Recombinant human RIPK2$^{1–300}$ and RIPK2$^{1–317}$ were produced using the baculovirus system in *sf21* insect cells. Cloning of DNA encoding RIPK2$^{1–300}$ with a tobacco etch virus protease (TEV) cleavable maltose-binding protein tag at the N-terminus in pFastBacHTB has been previously described (Pellegrini et al, 2017). Extension to residue 317 (RIPK2$^{1–317}$) of wild-type construct has been added by using the ABC (restriction/ligase)-free method (Qaidi & Hardwidge, 2019).

Plasmids pET49 encoding for human XIAP BIR2$^{154–240}$ and XIAP BIR2$_{AG}$$^{154–240}$ were a gift from Katrin Rittinger (Francis Crick Institute, UK). Both proteins comprise a PreScission protease (P3C) cleavable GST-tag at the N-terminus. Constructs XIAP BIR2$^{124–240}$, XIAP BIR2$_{AG}$$^{154–240}$, XIAP BIR2$^{124–263}$, XIAP BIR2$^{93–240}$, and XIAP BIR2$^{93–263}$ were cloned using the ABC method. The cloning resulted in two additional residues (GP) after the P3C cleavage site.

The sequences of the oligomers used for constructs extension are reported in Table S3.

## Protein expression and purification

N-terminally HIS-tagged TEV and N-terminally GST-tagged P3C proteases used for protein purification were produced at the Protein Expression and Purification Core Facility at EMBL.

RIPK2$^{1–300}$ and RIPK2$^{1–317}$ were expressed and purified using a similar protocol (Pellegrini et al, 2017). Bacmid generation, transfection, virus production, virus amplification, and protein expression have been performed at the EMBL Eukaryotic Expression Facility following the guidelines provided by the facility. Proteins have been expressed in *sf21* cells, using Sf-900 SFM medium (Gibco Life technologies). Cells were harvested 4 d post-infection and were lysed by sonication in buffer A (20 mM Tris pH 7.5, 300 mM NaCl, 50 mM NDSB, 5% glycerol, 500 $\mu$M TCEP) containing protease cocktail inhibitor (Complete, Roche). After centrifugation at 20,000$g$ for 30 min at 4°C, the supernatant solution was incubated for at least 2 h with amylose-affinity chromatography resin (New England Biolabs) while gently shaking at 4°C. The fusion protein was then eluted using the same lysis buffer supplemented with 40 mM maltose. Upon overnight TEV cleavage and dialysis against buffer B (20 mM Tris pH 7.5, 50 mM NaCl, 50 mM NDSB, 5% glycerol and 500 $\mu$M TCEP), proteins were further purified by anion exchange chromatography with a 0–1 M NaCl gradient, over two column volumes. Fractions corresponding to pure RIPK2 were then pooled and applied onto a PD-10 desalting column (GE Healthcare) equilibrated with buffer C (20 mM Tris pH 7.5, 150 mM NaCl, 2% glycerol). Proteins were then concentrated to 1–1.5 mg/ml and frozen in liquid N$_2$ for storage at –80°C.

For in vitro reconstitution of the complex and Native MS experiments, both RIPK2$^{1–300}$ and RIPK2$^{1–317}$ were used. For NMR and cryo-EM application, we exclusively used RIPK2$^{1–317}$.

XIAP BIR2 constructs were expressed in *E. coli* Rosetta 2 (Novagen) by growing the bacterial culture at 37°C until an OD$_{600\ nm}$ of 0.6 and inducing with 0.250 mM IPTG (isopropyl-$\beta$-D-1-

thiogalactopyranoside) overnight at 16°C. Cells were harvested and re-suspended in buffer A supplemented with 10 $\mu$M ZnCl$_2$ and protease inhibitor (Complete, Roche). Cells were then homogenized by sonication and the crude extract was centrifuged for 30 min at 18,000$g$. Clear lysate was incubated with glutathione sepharose resin (BRAND) for 2 h while gently shaking at 4°C. The beads were then washed with buffer A and the protein was eluted in buffer A supplemented with 20 mM reduced L-glutathione. Upon overnight P3C cleavage and dialysis against buffer D (20 mM Tris pH 7.5, 150 mM NaCl, 25 mM NDSB, 5% glycerol, 500 $\mu$M TCEP), the cleaved protein was separated from un-cleaved protein, GST, and GST-P3C by repeating the affinity step without incubation. The eluate was then aliquoted and frozen in liquid N$_2$ for storage at −80°C.

### Microscale Thermophoresis

As Tris is not compatible with primary amine fluorescent labelling, the buffer for all the samples (RIPK2$^{1-300}$, XIAP BIR2$^{154-240}$, XIAP BIR2$^{124-240}$, XIAP BIR2$^{93-240}$ and XIAP BIR2$^{93-263}$) was exchange by SEC in buffer E (20 mM Hepes pH 7.5, 150 mM NaCl, 2% glycerol). Labelling of RIPK2$^{1-300}$ was performed with Monolith Protein labelling kit RED-NHS second generation (MO-L011; Nanotemper) following the manufacturer's instructions. For labelling, the protein concentration was adjusted to be in the 6–10 $\mu$M range. For measurements, labelled RIPK2$^{1-300}$ was diluted to 40 nM, whereas the XIAP BIR2 constructs were titrated from 1.2–2,500 nM. 10 $\mu$l of each XIAP BIR2 constructs were mixed with 10 $\mu$l of labelled RIPK2$^{1-300}$ at a final concentration of 20 nM. Glass capillaries were filled with 4 $\mu$l of each mixture. Signal was improved by adding 0.05% Tween-20, by choosing red fluorescent excitation wavelength and by setting the LED power at 50%. All measurements were performed at 20°C.

### NMR

For NMR measurements, two samples were prepared: $^{15}$N, $^{13}$C-labelled XIAP BIR2$_{AG}^{124-240}$ and labelled XIAP BIR2$_{AG}^{124-240}$ in complex with RIPK2$^{1-317}$.

XIAP BIR2$_{AG}^{124-240}$ was expressed in *E. coli* Rosetta 2 (Novagen) by growing the bacterial culture at 37°C until an OD$_{600\ nm}$ of 0.6 and inducing with 0.500 mM IPTG for 5 h at 30°C in minimal growth medium M9 supplemented with either $^{15}$N-labeled NH$_4$Cl or a combination of $^{15}$NH$_4$Cl and $^{13}$C-labeled glucose. Labelled XIAP BIR2$_{AG}^{124-240}$ ($^{15}$N-XIAP BIR2$_{AG}^{124-240}$ and [$^{15}$N $^{13}$C] XIAP BIR2$_{AG}^{124-240}$) were purified as described above. After the second affinity step, samples were applied onto a Superdex 200 (10/300) increase (GE Healthcare) column equilibrated in buffer F (20 mM Tris pH 7.5, 150 mM NaCl). Labelled XIAP BIR2$_{AG}^{124-240}$ was directly concentrated on 3-kD molecular weight cut-off Centricon centrifugal filter units (EMD Millipore) until 300 $\mu$l at 240 $\mu$M were obtained.

RIPK2$^{1-317}$ buffer exchange in buffer F was done by using a PD-10 desalting column (GE healthcare). 5 mM MgCl$_2$ and 2 mM ATP were added to the kinase sample, and labelled XIAP BIR2$_{AG}^{124-240}$ was added in a ratio of 1.1:1 (RIPK2:XIAP BIR2). The sample was let on ice for 30 min and then concentrated on 3 kD molecular weight cut-off Centricon centrifugal filter units (EMD Millipore) until 300 $\mu$l at 200 $\mu$M were obtained.

All NMR experiments were performed at 25°C on Bruker Avance IIIHD spectrometers operating at magnetic field strengths of 850 or 950 MHz ($^1$H frequency), equipped with cryogenically cooled triple-resonance probes. $^1$H-$^{15}$N correlation spectra were recorded using a BEST-TROSY pulse scheme (Favier et al, 2011), whereas chemical shift assignments were obtained from a set of 3D BEST-TROSY HNC correlation experiments (Solyom et al, 2013). Translational diffusion constants of the proteins in solution were measured by 1D $^1$H DOSY experiments (Johnson, 1999). NMR data processing and analysis were performed using TopSpin 3.5 (Bruker BioSpin) and CCPNMR V3 software tools.

Heteronuclear {$^1$H}-$^{15}$N nOe (HETNOE) data were recorded as interleaved 2D data sets with and without $^1$H saturation before $^{15}$N excitation. The inter scan (recycle) delay was set to 5 s, and $^1$H saturation was applied for 3 s. The HETNOE values are computed as the peak intensity ratio measured in the $^1$H-saturated and reference spectra.

### Complex purification for native MS and cryo-EM

To remove residual GST and protein aggregates, a thawed XIAP BIR2 sample was applied onto a Superdex 200 (10/300) increase (GE Healthcare) column equilibrated in buffer C. In parallel, thawed RIPK2 was incubated with 500 $\mu$M AMPPCP and 5 mM MgCl$_2$ and let on ice for 30 min. Homogeneous XIAP BIR2 was then added in excess (ratio: 1:1.3) and complex (RIPK2$^{1-317}$ with either XIAP BIR2$^{154-240}$ or XIAP BIR2$_{AG}^{154-240}$) was let 30 min on ice. The sample was then concentrated 6 times by ultrafiltration using a 3-kD molecular weight cut-off Centricon centrifugal filter unit (EMD Millipore) and applied onto a Superdex 200 10/300 increase (GE Healthcare) in buffer C to remove excess XIAP BIR2. Homogeneity of the sample was then check by SDS–PAGE gel (Novagen), and the most concentrated complex fraction, usually in the concentration range of 5–7 $\mu$M, was used for cryo-EM grids preparation without further dilution. For Native MS experiments, fractions containing the complex were reapplied onto a Superdex 200 10/300 increase equilibrated in buffer G (250 mM ammonium acetate with 1 mM DTT) and measurements were immediately run afterwards.

### Native MS

RIPK2$^{1-300}$ or RIPK2$^{1-317}$ in complex with either XIAP BIR2$^{154-240}$ or XIAP BIR2$_{AG}^{154-240}$ were analysed by Native MS in the concentration range of 10 $\mu$M. Measurements were run as previously described (Pellegrini et al, 2017).

### Cryo-EM specimen preparation

Cryo-EM specimens were prepared on UltrAuFoil R1.2/1.3 holey gold grids (Quantifoil) that were glow-discharged for 20 s at 25 mA on both sides (PELCO easy glow). Vitrobot (Vitrobot Mk IV; Thermo Fisher Scientific) was set to 4°C and 100% humidity. A drop of 2 $\mu$l of the sample was applied to each grid side and blotting was run at force 0 for a total time of 3.5–4.5 s. Grids were then vitrified by plunging into liquid ethane at liquid N2 temperature. Grids were clipped into autoloader cartridges and screened using a Glacios cryo electron microscope (Thermo Fisher Scientific) equipped with

a Falcon 3 detector. Promising grids showing visible particles by eye at −1 μM defocus were used for data collection on CM01, ESRF Grenoble France (Fig S4C).

## Cryo-EM data collection and processing

The RIPK2$^{1–317}$-XIAP BIR2$^{154–240}$ dataset was acquired using a Titan Krios operating at 300 keV, equipped with K2 Quantum detector (Gatan) and a GIF Quantum energy filter (Gatan) at CM01 (ESRF) (Kandiah et al, 2019). 7,178 movies were collected in untilted mode, at 165K magnification, corresponding to a pixel size of 0.87 px/Å for a total dose of 47.8 (1.2 e$^{−/A2}$ per frame, fractionated in 40 frames). A total of 3,698 movies were recorded in tilted mode (1,870 micrographs at 20°, 1,825 micrographs at 25°), at the same magnification, for a total dose of 45.7 (1.1 e$^{−/A2}$ per frame, fractionated in 40 frames) (Fig S4D and Table S2).

Movies were imported into Relion, aligned and dose-weighted using MotionCor2 in Relion 3.1 (Fernandez-Leiro & Scheres, 2017; Zheng et al, 2017). Micrographs were then imported into Cryo-SPARC (Punjani et al, 2017) where CTF estimation has been run with Patch CTF and tilted micrographs were manually curated according to CTF-based estimated resolution (<10 Å) and estimated ice thickness (<1.1). Any attempts of processing the untilted dataset alone or in combination with the tilted datasets with a series of 2D and 3D classifications resulted in poorly resolved cryo-EM maps indicating preferred particle orientations. Therefore, we modified our processing approach. We firstly processed only the tilted data. The two tilted datasets (collected at 20° and 25° respectively) were kept separately and processed similarly (Fig S5). Particles were firstly picked on 100 representative micrographs using topaz (Bepler et al, 2019) and they were extracted with a box size of 300 × 300 pixels. We applied 2D classification with a tight mask (100 Å) and select 2D classes showing particle features. These particles were used to train topaz and the resulting model was used to pick particles in the remaining micrographs. After extraction, 2D classification was applied to eliminate bad particles and to select 2D class averages with lower background noise and stronger features. We then performed ab initio reconstruction with two models followed by non-uniform (NU) refinement using the best model as reference and all the particles. The resulting map denoted the tilted map, showing a better orientation distribution than for previous maps. To maintain a balance between the number of particles from the tilted dataset and the number of particles from the untilted dataset, only 2,000 micrographs of the latter were selected. Particles were then picked using as template projections of the tilted map. Resulting particles were extracted and submitted to 2D classification to eliminate bad particles. Remaining particles were then combined with tilted ones and submitted to another cycle of 2D classification. We then selected the particles with high effective classe-assigned values (ECA, between 2.1 and 2.6), as in our case, these were the ones showing the sharpest features and the lowest background noise. From the NU-3D refinement of these particles, we obtained the final map, at an average resolution of 3.15 Å (FSC 0.143 threshold). This map was used to calculate directional FSC and local resolution in CryoSPARC (Fig S6A–C). Adding more particles did not improve the map resolution and promoted

anisotropic resolution. Further 3D classification using either heterogeneous refinement or 3D classification (β) with different mask sizes did not improve particle selection nor the clarity of final map. A processing workflow, comprising data statistics, is shown in Fig S5.

For map sharpening, we used the sharpening tool in CryoSPARC. For manual reconstruction in Coot, the map was sharpened and blurred using the mrc_to_mtz tool in ccpem (Wood et al, 2015; Burnley et al, 2017).

## Model building and refinement

The atomic model of RIPK2$^{1–317}$ in complex with XIAP BIR2$^{154–240}$ was obtained by fitting with rigid-body refinement the existing crystal structures of both proteins. For RIPK2$^{1–317}$, we firstly fit the structure of active RIPK2 (PDB ID: 5NG0) (Pellegrini et al, 2017) which displays RIPK2 residues from 5 to 310. For XIAP BIR2, we used the crystal structure 4J3Y, chain C (Lukacs et al, 2013). We firstly fit the two structures in the cryo-EM map, using ChimeraX (Pettersen et al, 2021). After removal of waters, ions, and nucleotide analogue, we then proceeded with rigid-body refinement in Phenix (Liebschner et al, 2019). In Coot, we mutated back to cysteine residues 202 and 212 in the BIR2 domain, manually adjusted loop 210–214, built the αL-helices (residues 299–312, Fig 3D) using as template the structure of RIPK2–ponatininb (PDB ID: 4C8B) (Canning et al, 2015), and deleted the regions obviously outside of the density and the N-termini β-strand interaction. We then applied a second cycle of rigid-body refinement to obtain the final model. Figures were prepared with ChimeraX 1.4 (Pettersen et al, 2021). The software used in this project was installed and configured by SBGrid (Morin et al, 2013). Refinement statistics are reported in Table S2.

## Alphafold2

For the Alphafold2 calculations, we used Local ColabFold (Jumper et al, 2021; Evans et al, 2022 Preprint; Mirdita et al, 2022) installed on an in house server. To compute the complex structure of XIAP BIR2 with RIPK2 (residues 154–240, 1–317 respectively) we used three cycles, model "multimer-2" and 5 models. Neither increasing the number of cycles nor changing module improved the prediction. We applied the same calculation to RIPK2-cIAP1 BIR2 and RIPK2-cIAP2 (Fig S12C). As for the RIPK2–XIAP BIR2 complex, the prediction does not show any significant interaction between the two proteins.

## Mammalian cell culture and plasmids

A HEK293T cell line (from the laboratory of W. Filipowicz) was used, as previously described (Pellegrini et al, 2018). Cells were maintained in DMEM medium (Lonza) supplemented with 10% (vol/vol) FBS and nonessential amino acids (Gibco), at 37°C and 5% CO$_2$. Human XIAP full-length (1–497) construct for in cellulo experiments was purchased (puno1-hxiap; InvivoGen) and it was cloned from pUNO1 into the vector pcDNA3 using BamHI and XhoI restriction sites. Using the ABC method (Qaidi & Hardwidge, 2019) a P3C-cleavable MYC tag was added at the N-terminus together with a linker (ASASAS), resulting in pcDNA3-MYC-3c-XIAP(1–497). Used oligomers are reported in Table S3.

Single-amino acid mutants (RIPK2: R36L, D39L, R41L, R39L/R41L, N137L, N137L/N133L, K209A, K209R, E279L, S282L, K285A, and K285L; XIAP: Y176A, H128A, N209A, E211A, C213A, D214A, and R215A) were obtained by site-directed PCR mutagenesis of pcDNA3-HA-RIPK2 (1–540), and pcDNA3-MYC-3c-XIAP (1–497) using the oligos listed in Table S3.

### Immunoprecipitation and Western blot

For co-immunoprecipitation of MYC-XIAP, HA-RIPK2, and relative mutants, HEK293T cells were seeded in six-well plates, 24 h before transfection. Transfection was performed with LipoD293 transfection reagent (SignaGen). Each well was transfected with 275 ng of either pcDNA3-HA-RIPK2 or a corresponding mutant and 725 ng of either pcDNA3-Myc-3c-XIAP or a corresponding mutant. As negative control, a well was transfected with 1,000 ng of pcDNA3 empty vector. Each transfection mixture was completed with NOD2 activator MDP (tlrl-mdp; InvivoGen) and cells were lysed 22 h after transfection in 143 $\mu$l of cell lysis buffer (Cell Signaling) supplemented with protease inhibitor (Complete, Roche). According to the manufacturer's protocol, after 5 min of incubation on ice, cells were scratched from each well and sonicated briefly (two cycles of 7 s intercalated by incubation on ice). Residual cell debris were eliminated by centrifuging at 13,000$g$ for 15 min; 400 $\mu$l of a clean sample was then incubated overnight at 4°C on gentle shaking with 20 $\mu$l of either anti-MYC–agarose beads (3400; Cell Signaling) or anti-HA–agarose beads (3956; Cell Signaling). Beads were then washed five times with 500 $\mu$l of cell lysis buffer. To improve the quality of the immunoprecipitation experiment, the first wash lasted 30 min at 4°C on gentle shaking. Beads were finally resuspended with 20 $\mu$l 3X SDS sample buffer, and samples were loaded on 4–20% stain free SDS–PAGE prepacked gel (Bio-RAD), for immunoblot analysis. Rabbit anti-HA and mouse anti-MYC (3724S and 2276; Cell Signaling) were used for detection of transfected HA-RIPK2 or MYC-XIAP and corresponding mutants at 1:1,000 dilution. Rabbit anti $\beta$-actin antibody at 1:1,000 dilution was employed for normalization of total protein amount (8457; Cell Signaling). For revelation, secondary antibodies linked to fluorophores were used at 1:1,000 dilution: goat anti-rabbit and goat anti-mouse linked to Alexa 488 and Alexa 647, respectively (A11008 and A32728; Thermo Fisher Scientific).

### Production of RIPK2 mutants and binding to XIAP BIR2

Single-amino acid mutants (K209A, K209R, and S282L) were obtained by site-directed PCR mutagenesis of pFastBacHTB RIPK2 1–317, using the oligos listed in Table S3. RIPK2 mutants (RIPK2 K209A, K209R, and S282L) were expressed and purified following the same protocol of RIPK2 wt.

Each RIPK2 mutant was incubated with 500 $\mu$M ATP and 5 mM MgCl$_2$ and let on ice for 10 min. Homogeneous XIAP BIR2$^{154–240}$ was then added in excess (ratio: 1:2) and let further for 30 min on ice. The sample was then applied onto a Superdex 200 3.2/300 increase (GE Healthcare) equilibrated in buffer C. Complex formation was then check by SDS–PAGE gel (Novagen).

### RIPK2 reconstitution in RIPK2$^{-/-}$ iBMDMs

Gibson subcloning was performed to insert the human RIPK2 gene into the pCW57.1-Blast vector (#194067; Addgene) using oligos listed in Table S3. Site-directed mutagenesis was performed to generate the RIPK2 N137L allele using oligos reported in Table S3. HEK293T cells were then transfected with the pMD2 (#12259; Addgene), psPAX (#12260; Addgene), and indicated pCW57.1 construct at a ratio of 1 $\mu$g: 3 $\mu$g: 4 $\mu$g. 2 d after transfection, supernatant was harvested, centrifuged at 3,000$g$, and filtered through a 45-$\mu$m filter. RIPK2–/– immortalized bone marrow-derived macrophages (Chirieleison et al, 2016) were then transduced with the indicated virus in the presence of 10 $\mu$g/ml polybrene. 2 d after transduction, cells were exposed to 10 $\mu$g/ml blasticidin for 10 d before clones (>1,000) were pooled. Western blots were performed after exposure to 500 ng/ml doxycycline and showed identical levels of RIPK2 or RIPK2 N137L. qRT–PCR was then performed as described previously (Tigno-Aranjuez et al, 2013).

## Data Availability

Coordinates for RIPK2–XIAP BIR2 complex are in the protein data bank PDB with accession code 8AZA. The cryo-EM map has accession code EMD-15757. The NMR chemical shift assignments of XIAP BIR2$_{AG124–240}$ have been deposited with the Biological Magnetic Resonance Bank (BMRB) under accession number 51600.

## Supplementary Information

## Acknowledgements

We thank Katrin Rittinger (Francis Crick Institute, UK) for the kind gift of plasmids. We acknowledge Alice Aubert and Martine Pelosse for support in using the Eukaryotic Expression Facility at the EMBL in Grenoble. We acknowledge Sarah Schneider and Wojtek Galej for support in using the EM facility at the EMBL in Grenoble. We thank the European Synchrotron Radiation Facility for provision of beam time on CM01. We acknowledge the EM platform in HD for further data collections not here reported, but project connected. We thank the IT assistant of the IBS-EMBL shared processing cluster. This work used the platforms of the Grenoble Instruct-ERIC Center (ISBG; UAR 3518 CNRS-CEA-UGA-EMBL) within the Grenoble Partnership for Structural Biology (PSB), supported by FRISBI (ANR-10-INBS-0005-02) and GRAL, financed within the University Grenoble Alpes graduate school (Ecoles Universitaires de Recherche) CBH-EUR-GS (ANR-17-EURE-0003). Financial support from the IR INFRANALYTICS FR2054 CNRS and NIH Grant R35GM141603 are also acknowledged. We thank Caroline Mas for assistance and access to the biophysics platform. Molecular graphics and analyses were performed with UCSF ChimeraX, developed by the Resource for Biocomputing, Visualization, and Informatics at the University of California, San Francisco, with support from National Institutes of Health R01-GM129325 and the Office of Cyber Infrastructure and Computational Biology, National Institute of Allergy and Infectious Diseases.

## Author Contributions

M Lethier: data curation, validation, visualization, and methodology.
K Huard: validation, visualization, and methodology.
M Hons: methodology.
A Favier: data curation, formal analysis, validation, visualization, and methodology.
B Brutscher: data curation, formal analysis, validation, visualization, and methodology.
E Boeri Erba: data curation, formal analysis, validation, visualization, and methodology.
DW Abbott: data curation, validation, visualization, and methodology.
S Cusack: conceptualization, resources, supervision, funding acquisition, and writing—original draft, review, and editing.
E Pellegrini: conceptualization, resources, data curation, formal analysis, supervision, validation, investigation, visualization, methodology, project administration, and writing—original draft, review, and editing.

## Conflict of Interest Statement

The authors declare that they have no conflict of interest.

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
