## [Reviewer comments · Life Science Alliance]

Life Science Alliance

Structure shows that the BIR2 domain of E3 ligase XIAP binds across the RIPK2 kinase dimer interface

Mathilde Lethier, Karine Huard, Michael Hons, Adrien Favier, Bernhard Brutscher, Elisabetta Boeri Erba, Derek Abbott, Stephen Cusack, and Erika PELLEGRINI

DOI: <https://doi.org/10.26508/lsa.202201784>

Corresponding author(s): Erika PELLEGRINI, European Molecular Biology Laboratory and Stephen Cusack, European Molecular Biology Laboratory

Review Timeline:

Submission Date:	2022-10-24
Editorial Decision:	2022-11-15
Revision Received:	2023-07-04
Editorial Decision:	2023-07-16
Revision Received:	2023-08-09
Accepted:	2023-08-09

Scientific Editor: Novella Guidi

Transaction Report:

November 15, 2022

Re: Life Science Alliance manuscript #LSA-2022-01784-T

Erika Pellegrini
European Molecular Biology Laboratory
EMBL- Grenoble outstation
71 Avenue du Martyrs
Grenoble 38000
France

Dear Dr. Pellegrini,

Thank you for submitting your manuscript entitled "Structural analysis shows that the BIR2 domain of E3 ligase XIAP binds across the RIP2 kinase dimer interface" to Life Science Alliance. The manuscript was assessed by expert reviewers, whose comments are appended to this letter. We invite you to submit a revised manuscript addressing the Reviewer comments.

Thank you for this interesting contribution to Life Science Alliance. We are looking forward to receiving your revised manuscript.

Sincerely,

B. MANUSCRIPT ORGANIZATION AND FORMATTING:

Reviewer #1 (Comments to the Authors (Required)):

In this manuscript the authors examine binding of XIAP and RIP2K using several structural methods. They identify several residues on RIP2K that may be important for binding XIAP BIR2 domain. The authors also performed an overexpression binding assay in 293T cells to examine the binding of identified RIP2KL residues in cellular setting.

- The XIAP BIR2 AG designation needs to be better explained, does that come from the Sun et al, 1999 structure of XIAP BIR2?
- NMR data presented in Figure 2A support the formation of RIP2K-XIAP BIR2 complex. It also makes sense that only structural moieties that do not participate in complex formation remain detectable by NMR.
- However, the HETNOE data (Figure 2C) do not provide sufficient structural information because their data set lacks comparison with the bound state. In my opinion, these data do not enhance the conclusions made by the authors. Please add experimental details for this experiment in Methods.
- The authors claim that attenuated NMR peak intensities in the folded part are due to the relatively large particle size of the complex. It is not quite clear how they ruled out the possibility that the complex can be intrinsically heterogeneous.
- It would be important to make a distinction for which residues on RIP2K are important for homodimerization, which one for XIAP BIR2 binding and which contribute to both.
- The authors should perform NMR studies with RIP2K mutants (at least the critical ones like W170A from structural and binding studies) in complex with XIAP BIR2 to verify their data and hypothesis.
- The authors should produce recombinant RIP2K proteins for at least a few critical mutants and investigate their binding to XIAP BIR2 in a quantitative assay (biacore, SPR, FP, for example)
- Current cellular data are not sufficient to examine functional importance of RIP2K residues that were identified as critical for binding to XIAP BIR2. The authors should use NOD2 stimulated NF- κ B activation or cytokine release in cells expressing WT or RIP2K mutants that no longer bind XIAP BIR2 to verify their biological relevance.
- As the authors mention in their Abstract that their results have direct implications for RIP2K ubiquitination, they should verify if and how mutating identified RIP2K residues affects NOD2 stimulated RIP2K ubiquitination.

Reviewer #2 (Comments to the Authors (Required)):

Title: Structural analysis shows that the BIR2 domain of E3 ligase XIAP binds across the RIP2 kinase dimer interface

Manuscript # LSA-2022-01784-T

General Remarks

This study generates structural information for XIAP BIR2 binding to RIPK2. I am not a structural biologist so cannot comment on this aspect of it, however the data helps explain previous data and looks sound enough. I have only minor comments.

It has been impossible to predict the interaction between XIAP BIR2 and RIPK2, based on a classic IBM motif, even though the same interaction surface of BIR2 was clearly important, and as they show here a smac mimic can clearly compete. I think it would be useful if, now that the authors have defined the interaction, that they comment on the similarity of their motif to a classic IBM structurally?

Specific Remarks

page 4 - MacE et al, 2010

page 5 - see also Heim et al, 10.15252/embr.202050400; phosphorylation of RIPK2, even in unstimulated cells.

page 5: lack of role of RIPK2 kinase activity also in Nachbur et al, 10.1038/ncomms7442

page 6 for importance of linker - Silke et al, 10.1093/emboj/20.12.3114

page 9 prediction not predication

Reviewer #3 (Comments to the Authors (Required)):

In the current manuscript, Pellegrini and co-workers report the definitive model of the complex formation between the kinase domain of RIPK2 and XIAP BIR2 domain. RIPK2 is a key mediator of innate immune signaling downstream of NOD receptors, and its binding to XIAP BIR2 domain is a critical event in the signaling. The authors use NMR, mass spectrometry, cryo-EM and modeling to demonstrate that an anti-parallel homodimer of RIPK2 interacts with the BIR2 domain of XIAP. This interaction is established using two separate interfaces contributed by the N-lobe of one RIPK2 subunit and C-lobe of the second one. The authors also perform extensive mutagenesis/binding studies to validate the proposed mode of complex formation. Overall, this work provides a significant advancement in the understanding of RIPK2/XIAP binding and, therefore, signaling downstream from NOD receptors. I would like to recommend the manuscript for acceptance after the authors address several minor comments regarding data interpretation and presentation.

1. I should point out that many of the previous experiments utilizing mutant RIPK2 were performed using overexpression, which may easily obscure the fine details of signaling. While it is clear that RIPK2 can signal in the absence of catalytic activity (and without autophosphorylation in AS), it does not exclude some role of these autophosphorylation events under endogenous conditions. What is the phosphorylation status of AS in the RIPK2 molecules used for the analysis in the current paper? If the residues in AS are phosphorylated, could it contribute to the inability to see AS past residues 170-171? Autophosphorylation sites start at Ser174 according to the previous work by Pellegrini et al.
2. According to the current models, binding of kinase inhibitors may interfere with the ability of RIPK2 to assume conformation needed for XIAP binding. The authors should compare the XIAP binding interfaces on RIPK2 in the structures of inhibitor and BIR2-bound RIPK2 to see if there is indeed any clear difference that could specifically explain inhibition.
3. The role of K209 residues remains enigmatic. It is clearly important for signaling, but it may not serve as a poly-ub site, but rather play a role in the binding of RIPK2 to XIAP. However, could there be an explanation reconciling these two possibilities, i.e. could K209 serve to initially promote XIAP binding while its ubiquitination promotes subsequent XIAP dissociation, allowing ubiquitin chains to recruit TAK1 and LUBAC complexes?
4. Looking at the structure of the complex, salt bridge between Lys47 and Glu66 is absent in both subunits of RIPK2, indicating that RIPK2 is actually in an inactive GLu-out conformations, rather than in active Glu-in conformation.

Structural analysis shows that the BIR2 domain of E3 ligase XIAP binds across the RIPK2 kinase dimer interface

Life Science Alliance, answers to reviewers

Reviewer #1 (Comments to the Authors (Required)):

In this manuscript the authors examine binding of XIAP and RIPK2 using several structural methods. They identify several residues on RIPK2 that may be important for binding XIAP BIR2 domain. The authors also performed an overexpression binding assay in 293T cells to examine the binding of identified RIPK2L residues in cellular setting.

- The XIAP BIR2 AG designation needs to be better explained, does that come from the Sun et al, 1999 structure of XIAP BIR2?

The AG mutant comes from *Sun et al., 1999* where the authors modified the original construct to obtain a suitable sample for structure determination by NMR spectroscopy. The mutations C202A and C213G limit protein aggregation. Accordingly, we have clarified this point (page6, line 19).

- NMR data presented in Figure 2A support the formation of RIPK2-XIAP BIR2 complex. It also makes sense that only structural moieties that do not participate in complex formation remain detectable by NMR.

- However, the HETNOE data (Figure 2C) do not provide sufficient structural information because their data set lacks comparison with the bound state. In my opinion, these data do not enhance the conclusions made by the authors. Please add experimental details for this experiment in Methods.

The HETNOE data in figure 2C quantify the conformational flexibility in the N- and C-terminal regions of XIAP-BIR2 in its free form, underlining our conclusions that only these highly flexible protein segments remain visible in the complex. To better support this statement, we have modified the place where the figure is cited (page8-line 3).

As request, we have added further experimental details in the M&M section (page 19, line 9).

- The authors claim that attenuated NMR peak intensities in the folded part are due to the relatively large particle size of the complex. It is not quite clear how they ruled out the possibility that the complex can be intrinsically heterogeneous.

As stated on page 8, the disappearance (or strong attenuation) of peak intensities for the folded part of XIAP BIR2 indicates that these residues are involved in the formation of large NMR-invisible particles. The reviewer is right that the disappearance of NMR signals does not tell anything about the exact size (stoichiometry) or the homogeneity of this complex. Therefore we have performed additional translational diffusion NMR experiments that provide information about the apparent (average) size of the complex. Our NMR diffusion data are in good agreement with a 1:2 BIR2:RIP2 stoichiometry, although the presence of 2:2 BIR2:RIP2 complexes cannot be excluded (page 8, line 11).

- It would be important to make a distinction for which residues on RIPK2 are important for homodimerization, which one for XIAP BIR2 binding and which contribute to both.

The RIPK2-XIAP BIR2 structure shows that residues R36, R41, N133 and N137 are located at both protein-protein interaction and kinase dimerization interface, whilst the rest of the residues is contributing to RIPK2-XIAP BIR2 binding only. We have modified the paragraph entitled “ The RIPK2-XIAP interaction requires RIPK2 kinase dimerization” accordingly (pages 9-10).

- The authors should perform NMR studies with RIPK2 mutants (at least the critical ones like W170A from structural and binding studies) in complex with XIAP BIR2 to verify their data and hypothesis.
- The authors should produce recombinant RIPK2 proteins for at least a few critical mutants and investigate their binding to XIAP BIR2 in a quantitative assay (biacore, SPR, FP, for example)
- Current cellular data are not sufficient to examine the functional importance of RIPK2 residues that were identified as critical for binding to XIAP BIR2. The authors should use NOD2 stimulated NF- κ B activation or cytokine release in cells expressing WT or RIPK2 mutants that no longer bind XIAP BIR2 to verify their biological relevance.

In our study, we described the interaction between XIAP BIR2 and RIPK2. To validate the structure, we have mutated to either alanine or leucine several residues belonging to the interaction interface. The mutations were performed in tagged full length constructs, which were transiently expressed in mammalian cells. The cells were then stimulated for 22 h with NOD2 activator MDP, after which cells were harvested and pull-down were used to assess the effect of the mutations on complex formation (Fig. 4). The results obtained are either in agreement with published biochemical and *in cellulo* data (Hrdinka et al., 2018; Heim et al., 2020), or they describe new regions/residues involved in the RIPK2-XIAP BIR2 interaction, e.g. the α H-helix and residues N137, N133.

We have performed this kind of experiment, instead of expressing each kinase mutant in insect cells and XIAP BIR2 mutant in bacteria, for the following reasons:

- To be able to test multiple mutants simultaneously and in reasonable time.
- To avoid issues with expression and protein purification of unstable RIPK2 mutants. Our lab have had troubles in purifying RIPK2 mutants, in particular when residues are located at the kinase dimerization interface (Pellegrini E., et al. 2017, Plos One).
- To combine the construct transfection together with a plasmid encoding firefly luciferase under the control of NF- κ B promoter and test the effect of mutants on the NF- κ B activation by recording luciferase activity (as described in Pellegrini E., et al. 2018, Nat Comm).

Unfortunately, this last experiment never showed significant differences between control and mutants, most probably do to the presence of endogenous RIP2 and XIAP.

Based on the suggestions from the reviewer, we have expressed and purified several RIPK2 mutants and got soluble proteins for RIPK2-K209R, RIPK2-K209A, RIPK2-S282L. Before applying any quantitative assay, we have evaluated complex formation using size exclusion chromatography. Our SEC profiles show that K209R, K209A and S282L abolish binding completely, in agreement with our pull down from HEK cells (see supplementary figure 9). We then found reasonable to not apply further quantification assay.

In parallel, we have established a collaboration with Derek Abbott, who uses dox-inducible cell lines to test the effect of N137L RIPK2 mutants on genes expression, after NOD2 signaling induction. He confirmed that RIPK2 N137L reduces the expression of IRG1 and CXCL10 genes, both under the control of NF- κ B promoter.

In our paper we have also hypothesized a role for W170, which is not involved in the direct interaction with the BIR2 domain, but it appears always disordered in the crystallographic structure

of RIPK2 in complex with Ponatinib, GSK583 or CSLP18. Our pull down data showed that RIPK2 W170A mutants abort complex formation.

As requested, we expressed and purified RIPK2-W170A mutant and test binding with XIAP BIR2 using SEC. In parallel Derek Abbott engineered stable cell lines for RIPK2-W170A and tested the effect on NF- κ B genes. Both experiments show that RIPK2 W170A mutant retains the ability to bind XIAP BIR2 and the mutation does not alter the expression of neither IRG1 nor CXCL10 (Fig. 4 or 5).

A possible explanation of this discrepancy is that the original Western blot shows a strong fluorescent background, which decreases the signal itself and this might have led to a misinterpretation of the result. Consequently we have now removed the W170A observation and related data from the paper.

Paragraph “ The RIPK2-XIAP interaction requires RIPK2 kinase dimerization” (pages 9-10) and M&M section (pages 24-25) have been modified accordingly.

- As the authors mention in their Abstract that their results have direct implications for RIPK2 ubiquitination, they should verify if and how mutating identified RIPK2 residues affects NOD2 stimulated RIPK2 ubiquitination.

The abstract highlights the new findings revealed by our data, which focus on the scaffold role of RIPK2 kinase for XIAP BIR2 binding. With “direct implications for RIPK2 ubiquitination”, we meant the confirmation of the structural role of K209 and the improbability that it is a ubiquitination site. Therefore, we consider the proposed experiments beyond the scope of the current manuscript.

Reviewer #2 (Comments to the Authors (Required)):

Title: Structural analysis shows that the BIR2 domain of E3 ligase XIAP binds across the RIP2 kinase dimer interface

Manuscript # LSA-2022-01784-T

General Remarks

This study generates structural information for XIAP BIR2 binding to RIPK2. I am not a structural biologist so cannot comment on this aspect of it, however the data helps explain previous data and looks sound enough. I have only minor comments.

It has been impossible to predict the interaction between XIAP BIR2 and RIPK2, based on a classic IBM motif, even though the same interaction surface of BIR2 was clearly important, and as they show here a smac mimic can clearly compete. I think it would be useful if, now that the authors have defined the interaction, that they comment on the similarity of their motif to a classic IBM structurally?

We have added the following sentence at page 13, line 14:

“Indeed the BIR2 inhibitor engages polar contacts with residues of the XIAP loop 209-214, which we have here described to guide the interaction between the BIR2 domain and the C-lobe of Kinase_B.”

Specific Remarks

page 4 - MacE et al, 2010 , corrected to Mace et al

page 5 - see also Heim et al, 10.15252/embr.202050400; phosphorylation of RIPK2, even in unstimulated cells. Added accordingly at page 5 –line 22

page 5: lack of role of RIPK2 kinase activity also in Nachbur et al, 10.1038/ncomms7442, added

page 6 for importance of linker - Silke et al, 10.1093/emboj/20.12.3114 added

page 9 prediction not predication, corrected

Reviewer #3 (Comments to the Authors (Required)):

In the current manuscript, Pellegrini and co-workers report the definitive model of the complex formation between the kinase domain of RIPK2 and XIAP BIR2 domain. RIPK2 is a key mediator of innate immune signaling downstream of NOD receptors, and its binding to XIAP BIR2 domain is a critical event in the signaling. The authors use NMR, mass spectrometry, cryo-EM and modeling to demonstrate that an anti-parallel homodimer of RIPK2 interacts with the BIR2 domain of XIAP. This interaction is established using two separate interfaces contributed by the N-lobe of one RIPK2 subunit and C-lobe of the second one. The authors also perform extensive mutagenesis/binding studies to validate the proposed mode of complex formation. Overall, this work provides a significant advancement in the understanding of RIPK2/XIAP binding and, therefore, signaling downstream from NOD receptors. I would like to recommend the manuscript for acceptance after the authors address several minor comments regarding data interpretation and presentation.

1. I should point out that many of the previous experiments utilizing mutant RIPK2 were performed using overexpression, which may easily obscure the fine details of signaling. While it is clear that RIPK2 can signal in the absence of catalytic activity (and without autophosphorylation in AS), it does not exclude some role of these autophosphorylation events under endogenous conditions. What is the phosphorylation status of AS in the RIPK2 molecules used for the analysis in the current paper? If the residues in AS are phosphorylated, could it contribute to the inability to see AS past residues 170-171? Autophosphorylation sites start at Ser174 according to the previous work by Pellegrini et al.

The RIPK2 domain has been expressed and purified as described in Pellegrini et al, 2017. In that paper we published that RIPK2 was phosphorylated during insect cell expression and that we could stimulate the kinase autophosphorylation by adding further ATP-MgCl₂. To make the RIPK2-XIAP BIR2 complex, we added ATP-MgCl₂ to RIPK2 sample to be sure to have the kinase in the fully phosphorylated form. Therefore the AS in our structure is phosphorylated. Furthermore, in a recent unpublished crystal structure we have managed to visualize more of the disordered AS and see very clear density for P-Ser174 and P-Ser176. In this case the kinase was been purified in the same way but was co-crystallized with an inhibitor that promotes the inactive conformation (as described by the RIPK2 K47R structure in Pellegrini et al., 2017)

The adding of nucleotide to the complex is specified at page 8, line 19 in the main text, and at page 19, line 16 in M&M

The fact that the AS is disordered is not related to its phosphorylation state, but to the kinase being in the active conformation. Indeed, the AS is also disordered in the structure of RIPK2 D146N mutant, which is an un-phosphorylated mutant but in the active conformation (Pellegrini et al., 2017).

2. According to the current models, binding of kinase inhibitors may interfere with the ability of RIPK2 to assume conformation needed for XIAP binding. The authors should compare the XIAP binding interfaces on RIPK2 in the structures of inhibitor and BIR2-bound RIPK2 to see if there is indeed any clear difference that could specifically explain inhibition.

Ponatinib, GSK583 and CSLP18 are compounds that block XIAP BIR2 binding to RIPK2 (Goncharov et al., 2018; Hrdinka et al., 2018). We were expecting to see the reason by comparing their respective kinase-compound structure with the RIPK2-XIAP BIR2 structure. The only difference we found is that AS is completely disordered, even beyond residue 164.

See page 14, line 4

3. The role of K209 residues remains enigmatic. It is clearly important for signaling, but it may not serve as a poly-ub site, but rather play a role in the binding of RIPK2 to XIAP. However, could there be an explanation reconciling these two possibilities, i.e. could K209 serve to initially promote XIAP binding while its ubiquitination promotes subsequent XIAP dissociation, allowing ubiquitin chains to recruit TAK1 and LUBAC complexes?

It is an interesting hypothesis, but currently there is more data supporting the structural role of K209 rather than its possible role as ubiquitination site. Looking into actual literature, we cannot find any data positively confirming K209 as a ubiquitination site. The paper of Hasegawa et al., 2008 proposed K209 as a ubiquitination site because RIP2 with mutations K209A or K209R were not ubiquitinated. Recent mass spectrometry data does not report K209 as ubiquitination site (see Goncharov et al, 2018; Heim et al., 2020). Our structural and biochemical data clearly show that K209 has a structural role in XIAP BIR2 binding and that the mutations eliminate BIR2 binding.

4. Looking at the structure of the complex, salt bridge between Lys47 and Glu66 is absent in both subunits of RIPK2, indicating that RIPK2 is actually in an inactive Glu-out conformations, rather than in active Glu-in conformation.

In this manuscript we have defined the active or the inactive conformation of RIPK2 kinase, based on its C-helix position. The C-helix could be either in IN or OUT state, as described by the crystallographic structures of active RIPK2 and inactive RIPK2-K47R respectively (PDB Ids: 5NGO and 5NG3, Pellegrini E et al., 2017). Comparison between RIPK2-XIAP BIR2 and the cited crystallographic structures, show the C-helix to be in the IN position in the complex.

The IN position is supposed to promote the formation of a salt bridge between Glu66 and Lys47. However the density resolution is too low to correctly assign rotamers. Therefore we cannot exclude that binding of XIAP BIR2 to the RIP2K could induce a conformational change of the C-Helix which will break such bridge and perturb the kinase catalytic activity, activity which is not required to trigger the NF- κ B signaling.

July 16, 2023

RE: Life Science Alliance Manuscript #LSA-2022-01784-TR

Dr. Erika PELLEGRINI
European Molecular Biology Laboratory
EMBL- Grenoble outstation
71 Avenue du Martyrs
Grenoble 38000
France

Dear Dr. Pellegrini,

Thank you for submitting your revised manuscript entitled "Structure shows that the BIR2 domain of E3 ligase XIAP binds across the RIPK2 kinase dimer interface". We would be happy to publish your paper in Life Science Alliance pending final revisions necessary to meet our formatting guidelines.

- please address the final Reviewer 3's point
- please add the Twitter handle of your host institute/organization as well as your own or/and one of the authors in our system
- please note that the titles in the system and on the manuscript file must match
- please make sure the author order in your manuscript and our system match;
- please remove the count number of the abstract and summary blurb, and please remove the date when the paper has been resubmitted to LSA
- please make sure the manuscript sections are aligned with LSA's formatting guidelines: please separate the Figure legends and Supplemental Figure legends into separate sections
- please add callouts for Figures 3-D; 4A; S1A-D; S2A-B; S4A-b; S6A-C; S7A-D; S8A-; S9A-D; S11A-D; S12b to your main manuscript text

A. FINAL FILES:

B. MANUSCRIPT ORGANIZATION AND FORMATTING:

Sincerely,

Reviewer #1 (Comments to the Authors (Required)):

Authors have addressed all reviewers' questions and this manuscript can be published.

Reviewer #3 (Comments to the Authors (Required)):

The authors have adequately addressed my concerns with one minor issue. I think the explanation that is provided to my q4 is fine. However, because the salt bridge between Lys and Glu is a critical hallmark of Glu-in structure, the authors should provide the statement in the text saying that they assigned Glu-in conformation based on the location of the α C-helix, but low electron density does not allow the authors to confidently assign rotamers and confirm the presence of the salt bridge.

August 9, 2023

RE: Life Science Alliance Manuscript #LSA-2022-01784-TRR

Dr. Erika PELLEGRINI
European Molecular Biology Laboratory
EMBL- Grenoble outstation
71 Avenue du Martyrs
Grenoble 38000
France

Dear Dr. Pellegrini,

Thank you for submitting your Research Article entitled "Structure shows that the BIR2 domain of E3 ligase XIAP binds across the RIPK2 kinase dimer interface". It is a pleasure to let you know that your manuscript is now accepted for publication in Life Science Alliance. Congratulations on this interesting work.

DISTRIBUTION OF MATERIALS:

Again, congratulations on a very nice paper. I hope you found the review process to be constructive and are pleased with how the manuscript was handled editorially. We look forward to future exciting submissions from your lab.

Sincerely,
